# Increased L-Selectin on Monocytes Is Linked to the Autoantibody Profile in Systemic Sclerosis

**DOI:** 10.3390/ijms23042233

**Published:** 2022-02-17

**Authors:** Neža Brezovec, Katja Perdan-Pirkmajer, Tadeja Kuret, Blaž Burja, Snežna Sodin-Šemrl, Saša Čučnik, Katja Lakota

**Affiliations:** 1Department of Rheumatology, University Medical Centre Ljubljana, 1000 Ljubljana, Slovenia; katja.perdanpirkmajer@kclj.si (K.P.-P.); blaz.burja@kclj.si (B.B.); snezna.sodin@kclj.si (S.S.-Š.); sasa.cucnik@kclj.si (S.Č.); katja.lakota@kclj.si (K.L.); 2Faculty of Pharmacy, University of Ljubljana, 1000 Ljubljana, Slovenia; 3Faculty of Medicine, University of Ljubljana, 1000 Ljubljana, Slovenia; 4Institute of Cell Biology, Faculty of Medicine, University of Ljubljana, 1000 Ljubljana, Slovenia; tadejakuret@gmail.com; 5Faculty of Mathematics, Natural Sciences and Information Technologies, University of Primorska, 6000 Koper, Slovenia

**Keywords:** systemic sclerosis, monocytes, adhesion, chemotaxis, CD62L, CCR2, CCR5, CD11b, autoantibodies

## Abstract

Monocytes are known to be implicated in the pathogenesis of systemic sclerosis (SSc), as they exert prominent migratory, adhesive, and chemotactic properties. The aim of our study was to characterize the surface expression of adhesion/chemotactic molecules (CD62L, CD11b, CCR2, CCR5) on the SSc monocytes and determine correlations with the clinical presentation of SSc. We included 38 SSc patients and 36 healthy age-and sex-matched controls. Isolated monocytes, as well as in vitro serum-treated monocytes, were analyzed by flow cytometry; additionally, soluble CD62L was measured in serum. We found increased soluble CD62L in the SSc serum samples and increased CD62L on the surface of the SSc monocytes in the in the same set of patients. Among samples with determined SSc-specific autoantibodies, the surface CD62L was the lowest in patients positive for anti-PM/Scl autoantibodies and the highest in patients with anti-topoisomerase I autoantibodies (ATA). The treatment of isolated healthy monocytes with ATA-positive SSc serum resulted in increased surface CD62L expression. Moreover, surface CCR5 was reduced on the monocytes from SSc patients with interstitial lung disease but also, along with CCR2, negatively correlated with the use of analgesics/anti-inflammatory drugs and immunosuppressants. In conclusion, increased CD62L on SSc monocytes, particularly in ATA-positive patients, provides new insights into the pathogenesis of SSc and suggests CD62L as a potential therapeutic target.

## 1. Introduction

Systemic sclerosis (SSc) is an autoimmune disease characterized by the fibrosis of various organs, changes in the microvascular system, and dysregulation of the immune system. Autoantibodies such as anti-centromere (ACA), anti-topoisomerase I (ATA), and anti-PM/Scl are frequently present and associated with clinical symptoms. SSc is characterized by a heterogeneous clinical course, reflected by a broad range of organ involvement, including the skin, lungs, and gastrointestinal tract. Interstitial lung disease (ILD) is one of the major complications and the number one cause of mortality. Currently, there is no curative treatment for SSc, and all existing therapeutics are used to mitigate symptoms and limit disease progression [1]. Immunosuppressants and anti-inflammatory drugs are most commonly prescribed. Additionally, angiotensin-converting enzyme inhibitors, endothelin receptor antagonists, phosphodiesterase type 5 inhibitors, and prostacyclin analogs are prescribed for manifestations such as pulmonary arterial hypertension, vasculopathy, and renal crisis [2].

The pathogenesis of SSc is still not fully understood, but it is known that monocytes and monocyte-lineage cells are involved in a complex mosaic of cells and processes leading to disease development and progression [3]. In humans, circulating monocytes can be divided into three subsets based on the relative surface expression of CD14 and CD16, referred to as classical, intermediate, and non-classical subsets, with the latter also known as the inflammatory subset due to its strong pro-inflammatory activity [4]. Increased numbers of monocytes have been observed in SSc patients [5], and they have been proposed as a cellular biomarker for poor outcomes in fibrotic diseases [6]. In addition, activated non-classical monocytes in SSc produce increased levels of proinflammatory cytokines that may contribute to pulmonary fibrosis [7].

The altered surface expression of adhesion molecules on peripheral blood mononuclear cells (PBMCs) in SSc has been associated with clinical signs/symptoms and the use of immunosuppressive therapy [8]. In addition, SSc monocytes with altered phenotype and migratory activity contribute to the pathogenesis of ILD by entering tissues and transforming into other cell types such as macrophages and fibrocytes [9]. Monocyte-derived macrophages play a key role in the development of lung fibrosis [10] and in progressive SSc-ILD macrophage activation, and migration has been identified as one of the most enriched pathways [11]. Pro-inflammatory cytokines, such as tumor necrosis factor-α, interleukin (IL)-1, IL-6, and interferon-γ, which are frequently dysregulated in SSc, have been shown to increase adhesion molecules in the endothelium and contribute to endothelial dysfunction and vascular damage [12].

The migration of monocytes from circulation into the tissues follows a general cascade of events beginning with rolling along the endothelial surface, adhesion, and final transmigration. Various adhesion molecules, such as selectins and integrins, as well as chemotactic molecules, such as chemokines, are involved (Figure 1) [13].

L-selectin (CD62L), encoded by the gene *SELL*, has been characterized as a tethering/rolling receptor and is important for the transendothelial migration of leukocytes. During transmigration, enzymatic shedding of L-selectin occurs in the leukocytes, mainly by metalloproteinase domain-containing protein (ADAM17), resulting in a soluble L-selectin molecule whose function remains to be elucidated. Classical monocytes express higher levels of L-selectin, in contrast to intermediate and non-classical subsets [14]. 

CD11b together with CD18 forms a functional integrin known as Mac-1. It is expressed in leukocytes that, when it binds to its ligands (e.g., ICAM-1), mediates leukocyte rolling, stable adhesion, crawling, and transmigration through vessels into tissues. CD11b is also known to mediate phagocytosis and produce anti-inflammatory cytokines during inflammation when it binds to iC3b ligand [15]. In CD14+ SSc monocytes, genes related to adhesion were upregulated, in addition to exhibiting a decrease in CD52 and an increase in CD11b and CD18 at the protein level; the expression of adhesion molecules also correlated with enhanced monocyte adhesion [16].

Chemotaxis plays an important role in the redirection of monocytes into tissues through the binding of chemokine (C-C motif) ligands (CCL) to chemokine receptors (CCR), e.g., CCL2 (MCP-1) to CCR2, or CCL3 (MIP1α) to CCR5 [17,18]. CCR2 and CCR5 are most highly expressed in the classical monocytes, the subset that efficiently produces reactive oxygen species and differentiates into monocyte-derived macrophages or dendritic cells and as such, shape the pathological processes in tissues [19]. 

Chemokines, including CCL2, CCL3, and CCL5, have been found to be elevated in SSc patients [20]. 

However, to date, few studies have addressed the surface expression of adhesion/chemotactic molecules in SSc monocytes. The aim of our study is to investigate the surface expression of the adhesion molecules CD62L and CD11b and the chemotactic molecules CCR2 and CCR5 on the SSc monocytes. Secondly, we establish correlations with the clinical presentation of SSc and therapy, as well as demonstrate the causal effect of SSc serum on the surface expression of CD62L. Finally, we provide insight into the correlations between the monocyte surface expression of the adhesive/chemotactic molecules and pharmacotherapy in SSc.

## 2. Results

### 2.1. Characteristics of SSc Patients

We included 38 SSc patients and 36 age- and sex-matched healthy controls (HC) in our study (Table 1). The patients were predominantly female (82%) and their median age was 57 years. The clinical data of the SSc patients (Table 2) show that ATA were the most prevalent antibodies (45%), followed by ACA (37%), and PM/Scl (8%). The majority of SSc patients had limited cutaneous SSc (66%) and were classified as manifesting active disease stage based on the capillaroscopy pattern (55%). The median disease duration was 9 years. Of the more than 60% of patients with lung complications, 75% of them were diagnosed with ILD. One or more immunosuppressants (glucocorticoids, methotrexate, azathioprine, cyclophosphamide, mycophenolate mofetil, hydroxychloroquine, rituximab) were prescribed to 47% of patients, and analgesics/anti-inflammatory drugs (paracetamol, non-steroidal anti-inflammatory drugs, coxibs, metamizole, and opioids) to 34% of patients. Additional clinical and laboratory data of SSc patients are available in Appendix A.

### 2.2. Adhesion/Chemotactic Profile Is Altered on SSc Monocytes

To determine the adhesion and chemotactic profile of the SSc monocytes, we analyzed peripheral blood cells of SSc patients and HC by flow cytometry. Gating of monocytes and monocyte subsets was performed as shown in Figure 2. The surface expression of CD62L was significantly higher on the SSc monocytes compared to HC (median (IQR) MFI 27.6 (16.8) vs. 19.4 (13.3), *p* = 0.019), with the largest difference observed in the classical subset (MFI 33.5 (24.1) vs. 22.6 (16.4), *p* = 0.011). No differences in CD62L expression were observed in the intermediate subset, while CD62L expression was not detected in the non-classical subset. There was a significant decrease in CCR5 surface expression in the SSc intermediate monocytes compared to HC (MFI 5.5 (2.6) vs. 6.2 (2.4), *p* = 0.035) and a decrease in CD11b surface expression in the SSc total monocytes (MFI 26.1 (44.7) vs. 45.3 (36.2), *p* = 0.065), classical (MFI 31.9 (45.2) vs. 47.5 (36.4), *p* = 0.067), and non-classical subsets (MFI 5.4 (3.8) vs. 7.7 (9.7), *p* = 0.053) compared to HC. We observed no differences in the surface expression of CCR2 (Table 3).

### 2.3. The Surface Expression of CD62L on Monocytes Is Associated with the Autoantibody Profile of SSc Patients

To determine whether the surface expression of CD62L on the monocytes differs among SSc patients, we compared patients based on their autoantibody profile. The highest surface expression of CD62L on the monocytes was determined in ACA-, ATA-, or PM/Scl-negative patients (MFI 33.6 (13.3)), followed by ATA-positive (MFI 30.8 (20.3)) and ACA-positive patients (MFI 26.3 (11.1)). Interestingly, the CD62L MFI was significantly lower in PM/Scl-positive patients (MFI 6.2 (17.0)) than in the HC (MFI 19.4 (13.3)) (Figure 3a).

### 2.4. Soluble CD62L Is Increased in SSc Serum

During transmigration, the enzymatic shedding of L-selectin occurs in the leukocytes, releasing its soluble form and entering the circulation. Therefore, we measured the levels of soluble CD62L in SSc serum, collected at the same time as samples for flow cytometry. Soluble CD62L was significantly increased in the serum of SSc patients compared to the HC (median 1190 (451) ng/mL vs. 1128 (134) ng/mL, *p* = 0.0279) (Figure 3b). However, no correlation was found with the CD62L surface expression or autoantibody profile.

### 2.5. STreatment with SSc Serum Positive for Anti-Topoisomerase I Antibodies Increases CD62L on Monocytes

To see if the serum components of SSc patients can induce the surface expression of CD62L on the monocytes, we treated monocytes isolated from a healthy individual with 50% serum for 24 h. ATA-positive SSc serum increased the surface CD62L on the monocytes, although not significantly, compared to HC serum (MFI 3.8 (1.1) vs. 3.1 (0.7), *p* = 0.0571). Untreated monocytes served as the negative control (MFI 1.7) (Figure 3c).

### 2.6. The Expression of CCR5 on the Surface of Monocytes Is Associated with Interstitial Lung Disease

To discover clinically relevant information, we looked for correlations between the surface expression of adhesion/chemotactic molecules on the SSc monocytes and in the patients’ clinical data. The surface expression of CCR5 was significantly lower in SSc–ILD patients compared to SSc without ILD and HC in the total monocytes (MFI 2.7 (1.7) vs. 3.7 (1.4), *p* = 0.0277 and vs. 3.5 (2.1), *p* = 0.287), classical (MFI 3.2 (1.9) vs. 4.0 (1.5), *p* = 0.0237; and 3.7 (2.2), *p* = 0.0237), and intermediate subset (MFI 3.5 (2.5) vs. 5.9 (2.5), *p* = 0.0022; and 6.2 (2.4), *p* = 0.0017, respectively) (Figure 4).

### 2.7. The Surface Expression of CCR2 and CCR5 on SSc Monocytes Is Associated with Therapy

To determine whether pharmacotherapy could affect adhesion/chemotactic molecules on the surface of the monocytes, we compared surface expression based on the therapy that the SSc patients receive. We found that alterations in the surface expression of chemotactic molecules on monocytes were related to the use of analgesics/anti-inflammatory drugs and immunosuppressants (Figure 5). The surface expression of CCR5 was significantly reduced in the total monocytes of patients receiving methotrexate and rituximab compared to the non-treated patients (MFI 2.3 (1.6) vs. 3.2 (1.4), *p* = 0.0387 and 1.3 (0.9) vs. 3.3 (1.2), *p* < 0.0001), in the classical subset of patients on analgesics/anti-inflammatory drugs and rituximab (MFI 3.2 (1.5) vs. 3.6 (1.4), *p* = 0.0423 and 1.3 (1.0) vs. 3.6 (1.1), *p* < 0.0001), and in the intermediate subset of patients on immunosuppressants (MFI 4.0 (3.5) vs. 6.2 (2.9) *p* = 0.0229). The surface expression of CCR2 was significantly reduced in patients on analgesics/anti-inflammatory drugs compared to the non-treated patients in the total monocytes (MFI 13.0 (7.8) vs. 18.9 (13.6), *p* = 0.0399), classical subset (MFI 15.2 (6.6) vs. 22.0 (10.8), *p* = 0.0302), and in intermediate subset (MFI 1.6 (1.6) vs. 6.0 (8.3), *p* = 0.0171).

## 3. Discussion

While studies have previously addressed adhesion/chemotactic molecules in the leukocytes of SSc patients, our study is the first to report the surface expression of L-selectin (CD62L) on SSc monocytes. CD62L was significantly increased on the SSc monocytes, predominately in the classical monocyte subset (Table 3). As confirmed by our protein-level study, classical monocytes express the highest levels of CD62L, whereas intermediate and non-classical monocytes have much lower expression levels [21,22]. A microarray study of genes encoding adhesion molecules in the SSc monocytes revealed the upregulation of the gene *SELL* (encoding CD62L) in the SSc compared to the HC and suggested it as a candidate gene for further analysis [23].

Among samples with determined SSc-specific autoantibodies, ATA-positive patients had the highest surface expression of CD62L, followed by ACA- and PM/Scl-positive patients (Figure 3a). Similarly, a study by Sawaya et al. reported lower CD62L in ACA-positive compared to ACA-negative patients measured in PBMCs [8]. ATA are generally associated with diffuse cutaneous SSc with more severe organ damage [24], thus higher CD62L on monocytes from ATA-positive patients could contribute to a worse prognosis. In contrast, PM/Scl-positive patients show low CD62L surface expression on the monocytes usually develop limited cutaneous SSc with less-severe organ complications and a better prognosis [25]. The group of SSc patients who were negative for ATA, ACA, and PM/Scl and had the highest surface expression of CD62L were not tested for other SSc-specific autoantibodies, such as anti-RNA polymerase, anti-fibrillarin, and anti-Th/To antibodies, so no conclusions can be drawn for this group.

We also report increased serum levels of soluble CD62L in matched samples from SSc patients in whom higher levels of CD62L were measured on the monocytes (Figure 3b). Elevated levels have also been reported by other research groups [26,27] and were positively associated with inflammatory joint involvement, pitting scars, ulcers, and diffuse pigmentation [28]. In contrast, decreased serum CD62L has also been reported by other studies of SSc patients [29,30], with a negative correlation between soluble CD62L and disease activity or severity of diffuse cutaneous SSc [31], and similar levels have also been observed in the SSc and HC subjects [32,33]. Soluble CD62L can arise either from the splice variant transcript, which is more prevalent in rheumatic diseases, or from the classical shedding of surface CD62L via ADAM17 during transendothelial migration [14]. The gene and protein expression of ADAM17 was found to be increased in the CD14+ monocytes from patients with early SSc, but not in chronic SSc patients [34]. Moreover, it has been shown that reactive oxygen species, which also contribute to the pathogenesis of SSc, can trigger CD26L shedding in an autocrine-paracrine manner [31,32]. Increased soluble CD62L in SSc does not necessarily reflect CD62L shedding on monocytes, as serum levels also depend on the CD62L expression in other leukocytes. It is important to note that CD62L expression is higher in neutrophils, basophils, and NK cells than in PBMCs [21,22]. In PBMCs, CD62L has been found to be similar in the SSc and HC subjects [8], whereas opposing results have been reported for CD62L+ Tregs [35,36]. A lower surface expression of CD62L was found in the neutrophils [37] and NK cells [38] of SSc patients and in the CD3 T cells in SSc patients with PAH [39]; however, in those cell types, CD62L exhibited different functions than on monocytes. This indicates cell-type specific regulation of CD62L levels.

The surface expression of CD62L on monocytes treated with ATA-positive serum was increased in comparison to monocytes treated with HC serum, suggesting a causal relationship between SSc serum components (Figure 3c). The expression of CD62L is associated with an inflammatory microenvironment [40] and can be induced by cytokines such as interferon-gamma [41]. Moreover, ATA can induce a proinflammatory and proadhesive phenotype in fibroblasts by recognizing antigens on the fibroblast surface [42]. Further experiments are needed to confirm whether the observed effect depends on ATA, on other serum components of ATA-positive sera, or whether it also occurs with SSc serum containing other antibodies.

Monocyte-like THP-1 cell lines that do not express CD62L under basal conditions exhibit a more invasive phenotype when CD62L is present [43,44]. In bleomycin-induced fibrosis, a commonly used mouse model of SSc, deficiency of CD62L and/or ICAM-1 has been shown to suppress fibrosis in the skin and lungs and to reduce the number of macrophages in affected tissues among other leukocyte perturbances [45]. However, in the tight-skin mouse model of SSc, which lacks the typical inflammatory and vasculopathy features of SSc, CD62L deficiency did not inhibit the development of skin sclerosis [46]. Summarizing the findings, CD62L might be a good therapeutic target in the treatment of SSc.

Whereas decreased surface expression of CCR5 was found only in the intermediate subset of monocytes in all SSc patients (Table 3), significantly decreased CCR5 surface expression was found in all monocytes, classical monocytes, and intermediate monocytes from SSc patients with ILD (Figure 4). Accordingly, altered CCR5 was observed in several lung-related complications. It was reduced in lymphocytes in idiopathic pulmonary fibrosis [47], and the CCR5/prostaglandin D2 receptor CRTH2 ratio was also decreased in the T cells from SSc patients with ILD [48]. In contrast, increased CCR5 levels were observed in the lymphocytes and alveolar macrophages in sarcoidosis, but decreased with disease progression [49]. In addition, the elevated gene expression of CCR5 was found in intermediate monocytes from patients with chronic obstructive pulmonary disease as a result of high circulating IL-6 and sIL-6R levels [50].

Analgesic/anti-inflammatory drugs and immunosuppressants negatively correlated with the surface expression of the chemotactic molecules CCR2 and CCR5 on the SSc monocytes. CCR5 was most significantly reduced on both the total and classical monocytes from patients on rituximab, a monoclonal antibody directed against CD20 in B cells (Figure 5). The effect of rituximab on CCR5 has previously been reported in rheumatoid arthritis. After three months of rituximab therapy, the initial low surface expression of CCR5 in CD4-positive T cells increased in proportion to the decrease in disease activity, but the initial high CCR5 gene expression in PBMCs decreased to nearly physiological levels [51]. Reduced mRNA expression of CCR5 and CCR2 in PBMCs in RA patients has also been reported for diclofenac, a non-steroidal anti-inflammatory drug [52,53]. More than 80% of SSc patients with ILD involved in our study received analgesics/anti-inflammatory drugs and immunosuppressants, which correlated negatively with the surface expression of CCRs on the monocytes. Therefore, the use of analgesics/anti-inflammatory agents and immunosuppressants may partially explain the downregulated surface expression of CCR5 that we observed in SSc patients with ILD.

One drawback of our study was the high variability of the surface expression data for the adhesion/chemotactic molecules among patients, which may reflect the high heterogeneity of SSc patients in terms of different clinical presentations and disease duration. However, the limited number of patients did not allow us to consider all clinical manifestations. Interestingly, the CD62L surface expression on cultured monocytes was approximately 7-fold lower than that on freshly-isolated monocytes, suggesting that the conditions in culture might significantly alter the phenotype of monocytes. The low CD62L may be due to the static conditions in cell culture, in which monocytes are already adherent to the surface and do not adequately represent the in vivo dynamic conditions in circulation. We propose to use fluid assays in the future, which would also allow us to assess the functional consequences of increased monocyte CD62L.

In conclusion, our study demonstrated an altered surface expression of adhesion and chemotactic molecules on the SSc monocytes and monocyte subsets. CD62L, the most elevated molecule on the surface of SSc monocytes, correlated with the autoantibody profile. PM/Scl-positive patients had the lowest CD62L surface expression, followed by ACA-positive patients, while CD62L was highest in ATA-positive patients. However, further studies of monocyte treatment with isolated antibodies are needed to confirm the causative effect of ATA on CD62L. In addition, functional assays are necessary to define the contribution of CD62L to monocyte infiltration for considering CD62L as a potential therapeutic target for the treatment of SSc. Our study revealed decreased CCR5 surface expression on monocytes from SSc patients with ILD, but also showed that surface expression of CCR2 and CCR5 negatively correlated with analgesic/anti-inflammatory and immunosuppressant use. Therefore, the question of whether decreased CCR5 expression on monocytes might actually contribute to the detection of ILD, or if it is simply the result of a patients’ therapy still remains to be fully answered.

## 4. Materials and Methods

### 4.1. Patients and Controls

Consecutive patients (*n* = 38) who fulfilled the American College of Rheumatology classification criteria for SSc [54] were enrolled in this study, after providing informed consent, at the Department of Rheumatology, University Medical Centre Ljubljana. Patients were classified into sine scleroderma, limited cutaneous, or diffuse cutaneous subsets, according to the LeRoy criteria [55]. Clinical data were collected, including disease duration, presence of Raynaud’s phenomenon, digital ulcers, pitting scars, telangiectasia, calcinosis, and capillaroscopy results. Interstitial lung disease was diagnosed using high-resolution computed tomography (HRCT) or chest X-ray, and pulmonary arterial hypertension using right heart catheterization. Pulmonary function tests to determine the diffusing capacity for carbon monoxide (DLCO) and forced expiratory volume (FEV1) were performed. Blood tests were performed to measure the erythrocyte sedimentation rate (ESR), leukocytes, thrombocytes, haemoglobin, and serum amyloid A (SAA). To obtain serum, blood (3 mL) was collected from the SSc patients and HC subjects, allowed to clot undisturbed at room temperature for 30 min, and then centrifuged at 1800× *g* for 10 min at room temperature to separate serum, which was then collected, aliquoted, and frozen at −20 °C.

The presence of ATA and PM-Scl was determined in patients’ sera using the in-house method for anti-extractable nuclear antigens [56], whereas ACA were determined using an immunofluorescent HEp-2 assay, according to the manufacturer’s instructions (Immuno Concepts, Sacramento, CA, USA) and confirmed using the line immunoassay INNO-LIA ANA Update (Fujirebio, Tokyo, Japan), which is a useful tool for the detection and identification of ANA specificities in human serum. The patients’ treatment data were collected, including the use of analgesics, immunosuppressants, calcium channel blockers, prostacyclins, and phosphodiesterase type 5 inhibitors. Thirty-six age- and sex-matched HC subjects were randomly assigned for comparison.

### 4.2. Flow Cytometry

Peripheral blood samples from 31 patients and 36 age- and sex-matched HC subjects were collected in tubes (3 mL) containing heparin as an anticoagulant and processed the same day. Whole blood (100 µL) was incubated for 15 min at 4 °C with two different cocktails of fluorochrome-conjugated antibodies. The first cocktail contained anti-CD14-FITC (1:50, clone REA599, Miltenyi Biotec, Bergisch Gladbach, Germany), anti-CD16-PE (1:50, clone eBi-oCB16, eBioscience, Thermo Fisher Scientific, Carlsbad, CA, USA), anti-CD62L-PE -Vio770 (1:80, clone 145/15, Miltenyi Biotec, Bergisch Gladbach, Germany), and anti-CD11b-APC (1:40, clone ICRF44, eBioscience, Thermo Fisher Scientific, Carlsbad, CA, USA), while the second contained anti-CD14-FITC, anti-CD16-PE, anti-CCR2-APC (1:20, clone REA264, Miltenyi Biotec, Bergisch Gladbach, Germany) and anti-CCR5-PE -Vio770 (1:10, clone REA245, Miltenyi Biotec, Bergisch Gladbach, Germany). After washing twice with 1 ml of 0.5% BSA-FACS buffer, samples were lysed and fixed with whole blood lysing reagents (Beckman Coulter, Indianapolis, IN, USA) according to the manufacturer’s instructions, followed by second washing step and resuspension of the cells in FACS.

The serum-treated monocytes were washed twice with the FACS buffer, counted, checked for viability, and resuspended in FACS at a concentration of approximately 1 × 10^6^ cells/mL (approximately 2.5 × 10^5^ cells). The monocytes were then incubated for 15 min at 4 °C with anti-CD14-FITC (1:50, clone REA599, Miltenyi Biotec, Bergisch Gladbach, Germany), anti-CD16-PE (1:50, clone eBi-oCB16, eBioscience, Thermo Fisher Scientific, Carlsbad, CA, USA), and anti-CD62L-PE-Vio770 (1:80, clone 145/15, Miltenyi Biotec, Bergisch Gladbach, Germany), washed twice with FACS and resuspended in FACS.

Flow cytometry was performed using a MACSQuant Analyzer 10 (Miltenyi Biotec, Bergisch Gladbach, Germany), compensation was applied, and cell populations with corresponding data were analyzed using FlowJo (v10.5.3, Becton, Dickinson and Company, Ashland, OR, USA). Gating was performed as shown in Figure 2; the monocyte subsets were defined based on CD14 and CD16 as the classical (CD14++CD16−), intermediate (CD14++CD16+), and non-classical (CD14+CD16++) subsets, and the surface marker expression was presented as the median fluorescence intensity (MFI).

### 4.3. ELISA

Soluble L-selectin was measured in serum from the SSc patients and HC subjects using the Human sL-Selectin ELISA Kit (Invitrogen, Thermo Fisher Scientific, Carlsbad, CA, USA) following the manufacturer’s instructions.

### 4.4. Cell Experiment

The monocytes were isolated from the whole blood (8 mL) of a healthy individual. First, peripheral blood mononuclear cells (PBMCs) were isolated as previously described [57], followed by the isolation of monocytes by the indirect magnetic labeling of the monocytes using the Pan Monocyte Isolation Kit (Miltenyi Biotec, Bergisch Gladbach, Germany) according to the manufacturer’s protocol. Viability greater than 97% was determined using the Countess 3 Automated Cell Counter (Invitrogen, Thermo Fisher Scientific, Carlsbad, CA, USA), and successful separation of the monocytes was verified by flow cytometry using forward and side scatter parameters. The monocytes were resuspended in RPMI 1640 medium (Stemcell Technologies Inc., Vancouver, BC, Canada) in 24-well plates with a final concentration of 1 × 10^6^ cells/mL containing 50% heat-deactivated and sterile-filtered serum. This serum concentration approximates typical human blood conditions [58]. Eight different treatments were performed, four with serum from four ATA-positive SSc patients and four with serum from four age- and sex-matched HC subjects, in addition to one treatment without serum as a control. After incubation at 37 °C and 5% CO2 for 24 h, the cell culture medium containing the non-adherent monocytes was collected. The adherent monocytes were first detached with Accutase solution (Sigma-Aldrich Solutions, Merck, Darmstadt, Germany) before collection and resuspension in the FACS buffer, followed by a flow cytometry analysis.

### 4.5. Statistical Analysis and Clinical Significance

Normality of the Gaussian distribution was tested using the Shapiro–Wilk test. On this basis, statistical analysis of the data was performed using the Student *t*-test or the Mann–Whitney U test for comparison of two categorical variables, and the ANOVA or the Kruskal–Wallis test when more than two variables were compared. Values on the heat map were calculated as the log2 fold change of median MFI values in the monocytes from patients on a given therapy compared to values in the monocytes from patients who did not receive that respective therapy.

Graphpad Prism 9.3.0 software (GraphPad Software, San Diego, CA, USA) and IBM SPSS Statistics V22 (IBM, Armonk, NY, USA) were used to process the data. Data were expressed either as median or interquartile range; *p*-values of less than 0.05 were considered significant.

## Figures and Tables

**Figure 1 ijms-23-02233-f001:**
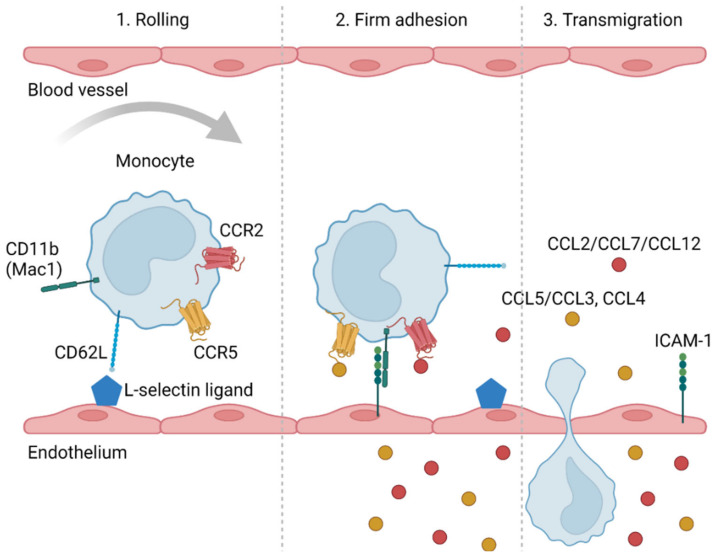
A schematic representation of the transendothelial migration of monocytes. Migration begins with the interaction of selectins (e.g., CD62L) with selectin ligands (e.g., CD34, GLYCAM1, MADCAM1). This results in the weak tethering and rolling of monocytes on the endothelial surface, followed by a firm adhesion mediated by integrins (e.g., Mac-1 (CD11b, CD18 complex)) to endothelial surface proteins (e.g., ICAM-1). The binding of chemokine ligands (e.g., CCL2, CCL7, CCL12, CCL5, CCL3, CCL4) to chemokine receptors (e.g., CCR2, CCR5) further enhances adhesion, enabling monocytes to transmigrate through the endothelium into the tissue. Figure 1 was created using BioRender.com (accessed on 25 January 2022).

**Figure 2 ijms-23-02233-f002:**
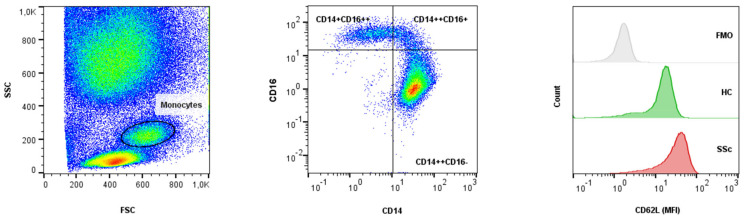
Gating of the monocytes based on forward and side scatter parameters, with the definition of the monocyte subsets, based on CD14 and CD16 surface molecules, as classical (CD14++CD16−), intermediate (CD14++CD16+), and non-classical (CD14+CD16++) [4]. The CD62L MFI shift to higher values is seen in the SSc monocytes compared to the HC. Negative control is represented as FMO. FMO—fluorescence minus one; FSC—forward scatter; HC—healthy controls; MFI—median fluorescence intensity; SSc—systemic sclerosis; SSC—side scatter.

**Figure 3 ijms-23-02233-f003:**
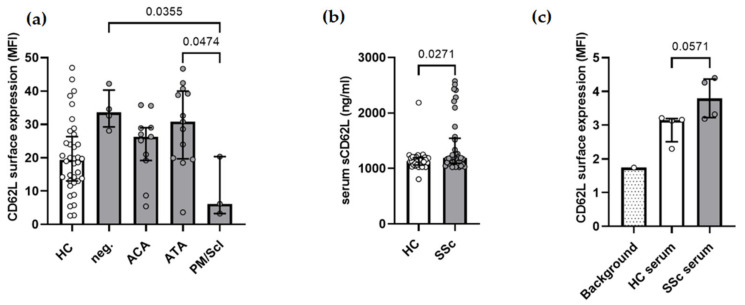
CD62L in SSc. (**a**) The surface expression of CD62L in SSc monocytes is associated with the autoantibody profile of the patients. CD62L is highest in patients negative for ACA, ATA, and PM/Scl, followed by ATA-positive patients and ACA-positive patients and the lowest in patients positive for PM/Scl antibodies. (**b**) Soluble CD62L in serum is significantly higher in the SSc patients than in the healthy controls. (**c**) Monocytes treated for 24 h with 50% serum from ATA-positive SSc patients (*n* = 4) express more surface CD62L than monocytes treated with serum from the healthy controls (*n* = 4). Values are shown as the median and interquartile range of median fluorescence intensity (MFI) (**a**,**c**) and the soluble serum CD62L concentration (**b**). ACA—anti-centromere antibodies; ATA—anti-topoisomerase I antibodies; HC—healthy controls; PM/Scl—anti-PM/Scl antibodies; SSc—systemic sclerosis.

**Figure 4 ijms-23-02233-f004:**
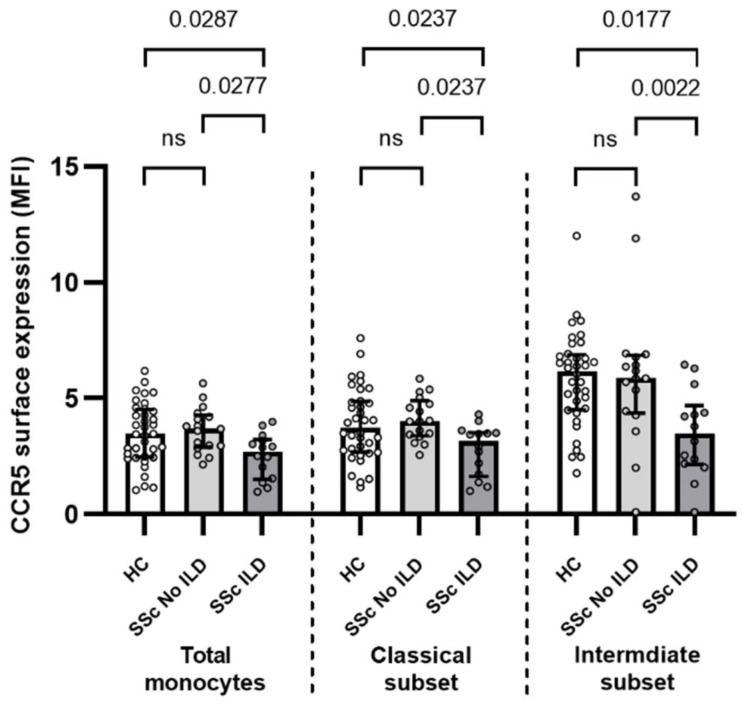
CCR5 surface expression in total monocytes, classical, and intermediate subsets is significantly lower in SSc patients with ILD compared with SSc patients without ILD and the healthy controls. HC—healthy controls; ILD—interstitial lung disease; SSc—systemic sclerosis.

**Figure 5 ijms-23-02233-f005:**
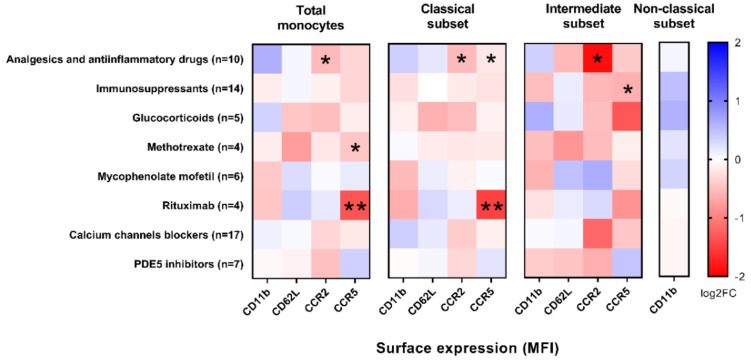
The surface expression of the chemotactic molecules CCR2 and CCR5 on the SSc monocytes and/or monocyte subsets is decreased in patients receiving analgesics/anti-inflammatory drugs and immunosuppressants (methotrexate and rituximab). Values are presented as log2FC (median MFI in treated SSc patients/median MFI in untreated SSc patients), with blue color representing positive associations and red color representing negative associations. Significance is indicated by * (*p* < 0.05) or ** (*p* < 0.01).

**Table 1 ijms-23-02233-t001:** The demographics of the SSc patients and the healthy controls.

Characteristics	SSc Patients	HC
Frequency *n* (%)	38 (51)	36 (49)
Age (years)		
Median (IQR)	57 (17)	55 (14)
Minimum	28	27
Maximum	88	84
Gender *n* (%)		
Male	7 (18)	5 (14)
Female	31 (82)	31 (86)

HC—healthy controls; IQR—interquartile range; SSc—systemic sclerosis.

**Table 2 ijms-23-02233-t002:** The clinical and laboratory data of the SSc patients.

Variable	*n* (%)
**Overlaps with other SARD**	
SSc only	30 (79)
Overlaps	8 (21)
**Clinical subset**	
Diffuse cutaneous	11 (29)
Limited cutaneous	25 (66)
Sine scleroderma	2 (5)
**Clinical manifestations**	
Raynaud’s	35 (92)
History of digital ulcers	21 (55)
Digital ulcers	7 (18)
Digital pitting scars	9 (24)
Telangiectasia	24 (63)
Calcinosis	8 (21)
GIT involvement	21 (55)
Lung involvement	24 (63)
Interstitial lung disease	18 (47)
PAH	1 (3)
**Autoantibodies**	
ACA	14 (37)
ATA	17 (45)
PM/Scl	3 (8)
Negative	4 (11)
**Capillaroscopy type**	
Early	2 (5)
Active	21 (55)
Late	5 (13)
No data	10 (26)
**Comorbidities**	
Hypertension	14 (37)
Hyperlipidemia	9 (24)
Diabetes II	1 (3)
Asthma	3 (8)
COPD	1 (3)
Cancer	3 (8)
CAD	3 (8)
Atherosclerosis	2 (5)
Other	19 (50)
**Treatment**	
Immunosuppressants	18 (47)
Glucocorticoids	7 (18)
Methotrexate	4 (11)
Azathioprine	1 (3)
Cyclophosphamide	2 (5)
Mycophenolate mofetil	8 (21)
Hydroxychloroquine	1 (3)
Rituximab	4 (11)
Analgesics and anti-inflammatory drugs	13 (34)
CCBs	21 (55)
Prostacyclins	2 (5)
PDE5 inhibitors	8 (21)

All values are presented as number and percentage of individuals. ACA—anti-centromere antibodies; ATA—anti-topoisomerase I antibodies; CAD—coronary artery disease; CCBs—calcium channel blockers; COPD—chronic obstructive pulmonary disease; GIT—gastrointestinal tract; PAH—pulmonary arterial hypertension; PDE5—phosphodiesterase type 5; SARD—systemic autoimmune rheumatic diseases; SSc—systemic sclerosis.

**Table 3 ijms-23-02233-t003:** The MFI of surface adhesion and chemotactic molecules on monocytes and monocyte subsets.

	Molecule		MFI Median (IQR)		*p*-Value
		Negative Control (FMO)	SSc Patients	HC	
**Monocytes**	CD62L	1.6	27.6 (16.8)	19.4 (13.3)	*0.019
	CD11b	0.7	26.1 (44.7)	45.3 (36.2)	0.065
	CCR2	0.2	17.1 (13.5)	18.7 (9.5)	0.211
	CCR5	0.4	3.1 (1.7)	3.5 (2.1)	0.302
**Classical subset**	CD62L		33.5 (24.1)	22.6 (16.4)	*0.011
	CD11b		31.9 (45.2)	47.5 (36.4)	0.067
	CCR2		20.1 (10.7)	20.9 (10.4)	0.172
	CCR5		3.6 (1.8)	3.7 (2.2)	0.443
**Intermediate subset**	CD62L		5.8 (9.6)	5.1 (3.1)	0.266
	CD11b		29.7 (40.8)	43.3 (28.1)	0.302
	CCR2		3.0 (6.8)	5.1 (6.5)	0.077
	CCR5		5.5 (2.6)	6.2 (2.4)	*0.035
**Non-classical subset**	CD62L		1.5 (1.3)	1.8 (1.5)	-
	CD11b		5.4 (3.8)	7.7 (9.7)	0.053
	CCR2		0.3 (0.1)	0.3 (0.1)	-
	CCR5		0.6 (0.3)	0.7 (0.2)	-

All values are presented as the median and interquartile range of median fluorescence intensity (MFI) for each molecule with respect to the total number of events detected by flow cytometry. *p*-value < 0.05 is considered significant (*). The MFI value of CD62L was significantly increased in the total SSc monocytes and in the classical monocyte subset, and the MFI value of CCR5 was decreased in the intermediate subset compared to HC. The MFI values of CD62L, CCR2, and CCR5 in the non-classical subset were similar to those in the negative control and were therefore excluded from further analyses. HC—healthy controls; IQR—interquartile range; FMO—fluorescence minus one; SS—systemic sclerosis.

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
