# Peer review of "Increased L-Selectin on Monocytes Is Linked to the Autoantibody Profile in Systemic Sclerosis"

_ijms, 2022, doi:10.3390/ijms23042233_

Round 1

Reviewer 1 Report

This work investigated the surface expression of adhesion molecules CD62L and CD11b and chemotactic molecules CCR2 and CCR5 on systemic sclerosis (SSc) monocytes. The study found increased soluble CD62L in SSc serum samples and increased CD62L on the surface of SSc monocytes in the same samples. This study also reports the contributory effect of SSc serum on the surface expression of CD62L. The study authors conclude that increased CD62L on SSc monocytes, especially in ATA-positive patients, provides new insights into pathogenesis of SSc and suggests CD62L as a potential therapeutic target.

The authors have done a nice job in writing this manuscript well. This work is of interest, but some concerns remain.

The abstract states that surface CD62L was highest in patients positive for anti-topoisomerase I autoantibodies (ATA).  However, this does not appear to be right. The authors themselves show that CD62L is highest in patients negative for ACA, ATA, and PM/Scl, followed by ATA-positive patients and ACA-positive patients (Figure 2a).  

Line 18: We found increased soluble CD62L in SSc serum samples and increased CD62L on the surface of SSc monocytes in the matched samples. Do the authors mean both soluble CD62L and surface CD62L in the same set of patient sample when they say matched samples?

Furthermore the authors state monocytes treated for 24 hours with 50% serum from ATA-positive SSc patients (n=4) express more surface CD62L than monocytes treated with serum from healthy controls. It is not clear in the manuscript whether they treated monocytes with ACA or PM/Scl positive sera to say that only ATA positive sera treated monocytes show increased expression of CD62L. In addition, it is not a clean experiment to incubate with whole sera and claim it is antibody mediated. The authors need to affinity purify ATA, ACA, or PM/Scl autoantibodies and incubate monocytes with a one of these purified antibodies and check for expression of CD62L. For control, the authors could incubate with equivalent control IgG for example from healthy controls or from Cohn fraction IgG.  

How CD62L soluble ligand measured in sera not given in methods section. 

How much peripheral blood collected from each subject? The authors state that serum-treated monocytes were resuspended in FACS buffer. Was this from patients and controls? However, it is not clear in the methods section how the authors isolated monocytes. How much blood was used from patients and controls for obtaining monocytes? 

Line 353: Serum was collected from SSc patients and HC subjects

Details of the assay for ACA, ATA, and PM-Scl needs to be given.

The authors do not provide pictures of ANA patterns for ACA, ATA, and PM-Scl

It is not serum that one collects from patients. Whole blood is collected and then serum separated and collected. Details of this needs to be given.

FACS staining conditions not given, nor is methodology given. How many cells were used for staining? What were the staining conditions used? 

How did the authors measure anti-centromere, anti-topoisomerase I, and anti-PM/Scl antibodies 

Minor comments 

PM/Scl needs to be expanded in the abstract 

Better not to start a sentence with a number. 

e.g., 38 consecutive patient 

Author Response

Author's Reply to the Review Report (Reviewer 1)

Comments and Suggestions for Authors

This work investigated the surface expression of adhesion molecules CD62L and CD11b and chemotactic molecules CCR2 and CCR5 on systemic sclerosis (SSc) monocytes. The study found increased soluble CD62L in SSc serum samples and increased CD62L on the surface of SSc monocytes in the same samples. This study also reports the contributory effect of SSc serum on the surface expression of CD62L. The study authors conclude that increased CD62L on SSc monocytes, especially in ATA-positive patients, provides new insights into pathogenesis of SSc and suggests CD62L as a potential therapeutic target.

The authors have done a nice job in writing this manuscript well. This work is of interest, but some concerns remain.

POINT1

The abstract states that surface CD62L was highest in patients positive for anti-topoisomerase I autoantibodies (ATA).  However, this does not appear to be right. The authors themselves show that CD62L is highest in patients negative for ACA, ATA, and PM/Scl, followed by ATA-positive patients and ACA-positive patients (Figure 2a).  

We appreciate the reviewer's observation. Our aim was to point out CD62L levels among samples positive for SSc specific autoantibodies. As samples in group “negative for ACA, ATA, and PM/Scl” might be heterogeneous, containing some other SSc specific autoantibodies (such as RNAPol II, Th/To and others) we did not want to comment this in abstract, but did mention this in Disscusion ( “The group of SSc patients who were negative for ATA, ACA, and PM/Scl and had the highest surface expression of CD62L were not tested for other SSc-specific autoantibodies such as anti-RNA polymerase, anti-fibrillarin, and anti-Th/To antibodies, so no conclusions can be drawn for this group.«)

To correct the sentence in the abstract we changed it as follows: “Among samples with determined SSc-specific autoantibodies, surface CD62L was lowest in patients positive for anti-PM/Scl autoantibodies and highest in patients with anti-topoisomerase I autoantibodies (ATA).”

We also corrected similar sentence in discussion: “Among samples with determined SSc-specific autoantibodies ATA-positive patients had the highest surface expression of CD62L, followed by ACA- and PM/Scl-positive patients…«

POINT2

Line 18: We found increased soluble CD62L in SSc serum samples and increased CD62L on the surface of SSc monocytes in the matched samples. Do the authors mean both soluble CD62L and surface CD62L in the same set of patient sample when they say matched samples?

We thank the reviewer for this comment. We changed the sentence as follows to make it more comprehensive: “We found increased soluble CD62L in SSc serum samples and increased CD62L on the surface of SSc monocytes in the in the same set of patients.”

POINT3

Furthermore, the authors state monocytes treated for 24 hours with 50% serum from ATA-positive SSc patients (n=4) express more surface CD62L than monocytes treated with serum from healthy controls. It is not clear in the manuscript whether they treated monocytes with ACA or PM/Scl positive sera to say that only ATA positive sera treated monocytes show increased expression of CD62L. In addition, it is not a clean experiment to incubate with whole sera and claim it is antibody mediated. The authors need to affinity purify ATA, ACA, or PM/Scl autoantibodies and incubate monocytes with a one of these purified antibodies and check for expression of CD62L. For control, the authors could incubate with equivalent control IgG for example from healthy controls or from Cohn fraction IgG.  

We appreciate reviewer consideration about treatment of monocytes with patients’ serum.

  1. a) When choosing group of samples for stimulating monocytes in in vitro study, we did not use ACA- or PM/Scl-positive sera (also stated in 4.4. Cell experiment section “Eight different treatments were performed, four with serum from four ATA-positive SSc patients and four with serum from four age- and sex-matched HC, in addition to one treatment without serum as a control.”) and we carefully reorganized one paragraph in Discussion not to claim we did comparisons:

“Surface expression of CD62L on monocytes treated with ATA-positive serum was in-creased in comparison to monocytes treated with HC serum, suggesting a causal relation of SSc serum components (Figure 3c). The expression of CD62L is associated with an inflammatory microenvironment [41] and can be induced by cytokines such as interferon-gamma [42]. Moreover, ATA can induce a proinflammatory and proadhesive phenotype in fibroblasts by recognizing antigens on the fibroblast surface [43]. Further experiments are needed to confirm whether the observed effect depends on ATA, on other serum components of ATA-positive sera, or whether it also occurs with SSc serum containing other antibodies.”

  1. b) We agree with the reviewer that it would be necessary to treat cells with purified antibodies to claim direct effect of antibodies. We are aware that ATA -positive SSc serum also contains other components (for example, interleukins (as described in Gourh et al. Arthritis Res Ther, 2009). However, because we had only a limited amount of serum available, it was unfortunately not possible to purify sufficient amount of antibodies from the samples used in this study. Nevertheless, we believe that our results still provide valuable information about the components of SSc serum that affect the adhesion properties of monocytes from a broader perspective.

Throughout the paper we emphasized the importance of all serum components (not only ATA autoantibodies) by using “ATA-positive SSc serum increased surface CD62L”. We specifically emphasized this also by adding “of SSc serum components” in line 276 “Surface expression of CD62L on monocytes treated with ATA-positive serum was increased in comparison to monocytes treated with HC serum, suggesting a causal relation of SSc serum components.”  We further point out the importance of treating with purified antibodies in the conclusion (line 332-334) as follows: “However, further studies of monocyte treatment with isolated antibodies are needed to confirm the causative effect of ATA on CD62L”.

POINT4

How CD62L soluble ligand measured in sera not given in methods section. 

Please see “4.3. ELISA” section for this information.

POINT5

How much peripheral blood collected from each subject?

We thank for this comment, the volume of collected blood was added in the text as follows:

“Serum was obtained from SSc patients and HC subjects. Blood was collected (3 mL) …”

“Peripheral blood samples from 31 patients and 36 age- and sex-matched HC were collected in tubes (3 mL) containing heparin …”

POINT6

The authors state that serum-treated monocytes were resuspended in FACS buffer. Was this from patients and controls?

Monocytes were isolated from whole blood of a healthy individual (Please see line 399). Monocytes were treated with serum (from patients and controls) and untreated (for negative control). Please see the section under 4.4. Cell experiment: “Eight different treatments were performed, four with serum from four ATA-positive SSc patients and four with serum from four age- and sex-matched HC, in addition to one treatment without serum as a control.”

POINT7

However, it is not clear in the methods section how the authors isolated monocytes. How much blood was used from patients and controls for obtaining monocytes? 

We thank the reviewer for this comment. Additional information was added regarding monocytes isolation in 4.4 Cell experiment section. “Monocytes were isolated from whole blood (8 ml) of a healthy individual. First, peripheral blood mononuclear cells (PBMCs) were isolated as previously described [58], followed by the isolation of monocytes by indirect magnetic labeling of monocytes using the Pan Monocyte Isolation Kit (Miltenyi Biotec, Bergisch Gladbach, Germany) according to the manufacturer’s protocol.”

POINT8

Line 353: Serum was collected from SSc patients and HC subjects

It is not serum that one collects from patients. Whole blood is collected and then serum separated and collected. Details of this needs to be given.

We appreciate this comment, we corrected the sentence and added the information.

“To obtain serum, blood (3 mL) was collected from SSc patients and HC subjects, allowed to clot undisturbed at room temperature for 30 minutes, and then centrifuged at 1800 x g for 10 minutes at room temperature to separate serum, which was then collected, aliquoted and frozen at -20°C.”

POINT9

Details of the assay for ACA, ATA, and PM-Scl needs to be given.

How did the authors measure anti-centromere, anti-topoisomerase I, and anti-PM/Scl antibodies 

ACA, ATA, and PM-Scl autoantibodies were measured as part of routine laboratory checkup and not specifically for this study. ATA and PM-Scl were determined using the in-house method for anti-extractable nuclear antigens (which is described in detail in reference 57), whereas ACA were determined using a commercial assay (Immuno Concepts, Sacramento, USA).

We have changed the description of the determination in Section 4.1 : "The presence of ATA and PM-Scl was determined in patients' sera using the in-house method for anti-extractable nuclear antigens [57], whereas ACA were determined using an immunofluorescent HEp-2 assay according to the manufacturer's instructions (Immuno Concepts, Sacramento, USA) and confirmed using the line immunoassay INNO-LIA ANA Update (Fujirebio, Tokyo, Japan), which is a useful tool for the detection and identification of ANA specificities in human serum."

POINT10

The authors do not provide pictures of ANA patterns for ACA, ATA, and PM-Scl

Because the determination of the autoantibody profile was part of the clinical/laboratory information about the patients collected for this study and was not made specifically for this study, we did not consider these pictures to be an essential part of our manuscript. However, we can provide ANA patterns for these three autoantibodies if the reviewer considers it crucial for the interpretation of our data. However, we would like to point out that ANA pattern for ACA is a specific pattern, whereas ATA and PM-Scl are not antigen-specific immunofluorescence patterns, so the autoantibodies need to be confirmed by another method, which was done using the in-house method for anti-extractable nuclear antigens.

POINT11

FACS staining conditions not given, nor is methodology given. How many cells were used for staining? What were the staining conditions used? 

We agree with that observation and we added the methodology in the paragraph as follows: “Serum-treated monocytes were washed twice with FACS buffer, counted, checked for viability, and resuspended in FACS at a concentration of approximately 1 x 106 cells/mL (approximately 2.5 x 105 cells). Monocytes were then incubated for 15 minutes at 4°C with anti-CD14-FITC (1:50, clone REA599, Miltenyi Biotec, Bergisch Gladbach, Germany), anti-CD16-PE (1:50, clone eBi-oCB16, eBioscience, ThermoFisher Scientific, Carlsbad, CA, USA), and anti-CD62L-PE-Vio770 (1:80, clone 145/15, Miltenyi Biotec, Bergisch Gladbach, Germany), washed twice with FACS and resuspended in FACS.”

Minor comments 

POINT12

PM/Scl needs to be expanded in the abstract 

We thank the reviewer for this comment. However, PM/Scl term is commonly used and not expanded in papers. See the references below.

Rituximab and intravenous immunoglobulin treatment in PM/Scl antibody-associated disease: case-based review (Rheumatol Int. 2022)

The prevalence and significance of anti-PM/Scl antibodies in systemic sclerosis (Ann Agric Environ Med. 2021)

PM-Scl and Th/To in systemic sclerosis: a comparison of different autoantibody assays (Clin Rheumatol. 2021)

POINT13

Better not to start a sentence with a number. 

e.g., 38 consecutive patient 

We appreciate this comment and we corrected the sentences as follows: “Consecutive patients (n=38)…”

Reviewer 2 Report

The authors gave interesting overview about surface expression of adhesion/chemotactic molecules (CD62L, CD11b, CCR2, CCR5) on SSc monocytes and determined correlations with the clinical presentation of SSc.

However, a few corrections should be made:

Title: Increased L-selectin on Monocytes is Linked to Autoantibody Profile in SSc”

Please don’t write abbreviation in the title: “SSc”. Insert a whole words.

Abstract: First sentence is a little bit awkward: „Monocytes have previously been implicated in the pathogenesis of systemic sclerosis (SSc) as they exert prominent migratory, adhesive, and chemotactic properties.”

And they are not implicated now? Please rephrase it.

Figures: Increase a resolution of Figure 2 and Figure 5.

Author Response

Author's Reply to the Review Report (Reviewer 2)

The authors gave interesting overview about surface expression of adhesion/chemotactic molecules (CD62L, CD11b, CCR2, CCR5) on SSc monocytes and determined correlations with the clinical presentation of SSc.

However, a few corrections should be made:

POINT1:

Title: „Increased L-selectin on Monocytes is Linked to Autoantibody Profile in SSc”

Please don’t write abbreviation in the title: “SSc”. Insert a whole words.

We appreciate the reviewer's comment and agree with it, title is now changed to „Increased L-selectin on Monocytes is Linked to Autoantibody Profile in Systemic Sclerosis” (see track changes).

POINT2:

Abstract: First sentence is a little bit awkward: „Monocytes have previously been implicated in the pathogenesis of systemic sclerosis (SSc) as they exert prominent migratory, adhesive, and chemotactic properties.”

And they are not implicated now? Please rephrase it.

We thank the reviewer for pointing out the unclear message in the sentence. We have rephrased it as follows: “Monocytes are known to be implicated in the pathogenesis of systemic sclerosis (SSc) as they exert prominent migratory, adhesive, and chemotactic properties” (see track changes).

POINT3:

Figures: Increase a resolution of Figure 2 and Figure 5.

We are grateful for this observation. The quality has been increased for both Figures to the maximum.

Reviewer 3 Report

The npaper is interesting and well written.

I suggest to improve the paper by discussing the role of VEGF in accelarated atherosclerosis (see and add as references paper by Colombo et al concerning intima media thickness in SLE and the papers by Murdaca et al concerning  VEGF and free radicals in endothelial dysfunction).

Author Response

Author's Reply to the Review Report (Reviewer 3)

Comments and Suggestions for Authors

The npaper is interesting and well written.

POINT1:

I suggest to improve the paper by discussing the role of VEGF in accelarated atherosclerosis (see and add as references paper by Colombo et al concerning intima media thickness in SLE and the papers by Murdaca et al concerning  VEGF and free radicals in endothelial dysfunction).

We thank the reviewer and appreciate his suggestion to improve our paper. We have added a valuable sentence about adhesion molecules contributing to endothelial dysfunction, citing the review by Murdaca et al. from 2013: "Pro-inflammatory cytokines such as tumor necrosis factor-α, interleukin (IL)-1, IL-6, and interferon-γ, which are frequently dysregulated in SSc, have been shown to increase adhesion molecules in the endothelium and contribute to endothelial dysfunction and vascular damage." Unfortunately, we could not include the interesting data on intima media thickness in SLE, because our paper focuses primarily on systemic sclerosis and monocyte migration and adding this information is beyond the scope of our paper.

Round 2

Reviewer 1 Report

Dear Editors,

Tha authors have addressed most of my comments

Sincerely,

Biji Kurien